# Using Embeddings for Causal Estimation of Peer Influence in Social Networks

Irina Cristali[1]     Victor Veitch[1,2]

[1]*Department of Statistics, The University of Chicago*
[2]*Google Research*

## Abstract

We address the problem of using observational data to estimate peer contagion effects, the influence of treatments applied to individuals in a network on the outcomes of their neighbors. A main challenge to such estimation is that *homophily*—the tendency of connected units to share similar latent traits—acts as an unobserved confounder for contagion effects. Informally, it's hard to tell whether your friends have similar outcomes because they were influenced by your treatment, or whether it's due to some common trait that caused you to be friends in the first place. Because these common causes are not usually directly observed, they cannot be simply adjusted for. We describe an approach to perform the required adjustment using node embeddings learned from the network itself. The main aim is to perform this adjustment nonparametrically, without functional form assumptions on either the process that generated the network or the treatment assignment and outcome processes. The key contributions are to nonparametrically formalize the causal effect in a way that accounts for homophily, and to show how embedding methods can be used to identify and estimate this effect. Code is available at https://github.com/IrinaCristali/Peer-Contagion-on-Networks.

## 1   Introduction

We are interested in estimating peer effects, the causal influence of individuals on their neighbors.

**Example.** We want to infer the effect that social pressure has on vaccination. Suppose we observe data from a population where each unit $i$ is a person in an interconnected social network, and for each unit we know whether they were vaccinated at the beginning of the study period, $T_i$, and whether they were vaccinated at the end of the study period, $Y_i$. We are interested in estimating the effects of the treatment $T_i$ of person $i$ on the outcome $Y_j$ of person $j$. In addition to their vaccination status, each unit has attributes $C_i$ that act as (proxies for) causes of both the particular network ties they form, and their vaccination behavior. For instance, $C_i$ may include age, race, education status, income level, political affiliation, and so forth.

The core challenge here is that we want to estimate a causal effect (e.g., what would happen if we intervened by vaccinating popular people?), but the variables $C_i, C_j$ act as confounders between the treatment $T_j$ and outcome $Y_i$. The reason is that when defining a contagion effect from $j$ to $i$ we must condition on the presence of an edge between $(i, j)$. The edge is causally influenced by both $C_i$ and $C_j$, so the conditioning creates a dependency between these variables (it acts as a collider). For example, if we learn that Alice and Bob are friends then we can infer that they likely live in the same city. Accordingly, the association between $T_j$ and $Y_i$ may be either due to *contagion*: the causal influence of $T_j$ on $Y_i$ (Bob got vaccinated because Alice did), or due to *homophily*: the tendency of similar people to be connected in a network (e.g., Alice and Bob got vaccinated because they both

live in a major city, and it's only due to chance that Alice got vaccinated before Bob). In general, homophily is confounded with contagion [ST11].

Now, if we observed the attributes $C$, we could formalize the contagion effect using standard causal tools and then identify the effect from observational data by adjusting for $C$ [EKB16; EB17; BDF20]. However, often such detailed knowledge of $C$ is unavailable. In this case straightforward causal identification and estimation is not possible.

In such situations, we might hope to make use of the following intuitive observation. The pattern of network ties itself carries information about each $C_i$. Indeed, this is the root problem that causes the confounding issue. Accordingly, we might hope to use the network itself to obtain estimates $\hat{C}_i$ for the attributes of each node. Then, we could adjust for the estimated $\hat{C}_i$ in some suitable causal estimation procedure. The aim of this paper is to clarify this type of procedure.

Shalizi and McFowland III [SM16] provide an estimation strategy of this kind. They first assume that the network is generated by either a stochastic block model [HLL83; LW19] or a latent space model [HRH02; RFR16]. They assume the attributes $C$ correspond to the latent community identities or latent space positions. They then further assume that the outcome of each unit is defined by a particular linear structural equation, which includes a term for both the average treatment of that node's neighbors, and a term for the effect of the attributes $C_i$. Their procedure is to estimate $C$ using the assumed network model, and then use the estimated $\hat{C}$ in a linear regression to determine the coefficient of the average-neighbor-treatment term.

This approach, however, critically relies on the assumed parametric form of both the network model and the outcome model. Indeed, even the target of estimation is defined as a parameter of this assumed model. Accordingly, when the parametric network model or linear outcome model is misspecified, the meaning of both the estimand and estimation procedure becomes unclear.

The goal of this paper is to develop a non-parametric procedure in this vein. That is, the aim is causal estimation of contagion effects by adjusting for network-inferred attributes without relying on detailed parametric assumptions. Towards this end, the paper follows three main steps:

1. Formalize the target causal effect non-parametrically. The main challenge is that the estimand must depend on the network we are working with (because contagion requires knowing who is friends with whom) and the network must itself be modeled as a random variable which is a function of the unobserved confounders $C$ (to accommodate homophily).

2. Derive sufficient conditions for the estimated attributes to yield causal identification. The idea is that it is not necessary to exactly reconstruct $C$, but only extract the minimum information that will identify the causal effect—this turns out to be (plausibly) a much easier task.

3. Give a concrete method for contagion estimation using node embedding techniques to extract the information from the network that is relevant for predicting peer influence, and illustrate the practical performance of this technique.

## 2   Setup

Consider a network $G_n$ of $n$ individuals, where connections between people are encoded through undirected edges between nodes. We define the degree of a node as the number of connections it has. The neighbors of node $i$ are the nodes with which $i$ has ties. Each such link is captured by the network adjacency matrix $A$, where $A_{ij} = \mathbb{1}_{\{i \text{ and } j \text{ share a tie}\}}$. We take $A_{ii} = 0$ for all $i$. We also consider a vector of variables associated with each node: $(Y_i, C_i, T_i)$. Here, $Y_i$ is the observed outcome, $T_i$ is the treatment, and $C_i$ are (unobserved) attributes that may causally influence $Y$, $T$ and $A$.

The data is generated according to the following structural equation model, adapted from Ogburn et al. [Ogb+17] to model the network as a random variable.[1]

---

[1]We simplify by removing the influence of neighbors' covariates on outcomes and treatments.

$$\begin{aligned}
C_i &\leftarrow f_C[\varepsilon_{C_i}]; \\
A_{ij} &\leftarrow f_A[\{C_i\}_i, \varepsilon_{ij}]; \\
T_i &\leftarrow f_T[C_i, \varepsilon_{T_i}]; \\
Y_i &\leftarrow f_Y[S_Y(\{T_j : A_{ij} = 1\}), C_i, \varepsilon_{Y_i}]
\end{aligned} \tag{2.1}$$

The $\varepsilon$ variables represent exogenous noise, which we take to be identically distributed and independent of the network and of each other. The function $S_Y$ summarizes the neighbors' treatment—e.g., $S_Y$ could be the average function, or the logical OR function. This means that the treatment assignment for each node $i$ depends only on its attributes $C_i$, the outcome assignment depends on the node attributes $C_i$ and the treatments of all its neighbors, and the network structure depends on all $C$. The structural functions $f$ are fixed but unknown.

## 3 Formalizing the causal estimand

Consider setting $T_i \leftarrow t_i^*$ for each unit $i$. Our aim is to formalize the idea of "average influence of a node's neighbors' treatments on its outcome".

An intuitive choice is

$$\psi_{t^*}^{\text{full info}} := \frac{1}{n} \sum_{i=1}^{n} \mathbb{E}[Y_i | \mathrm{do}(T = t^*), \{C_i\}_i, G_n]. \tag{3.1}$$

Here, $\mathrm{do}$ is Pearl's do notation, and indicates that the treatment (vector) of all nodes is intervened on and set to $t^*$. Note that, following the structural model in equation 2.1, nodes' outcomes are only affected by their neighbor's treatments. Accordingly, the estimand is the average outcome we would have seen had the treatment assignment been set to $T \leftarrow t^*$, keeping both the node attributes $\{C_i\}$ and the network $G_n$ fixed. The interpretation of this effect is: the average outcome under the hypothetical treatment, applied to the *same set of people* connected by the *same link structure*.

Unfortunately, this estimand will not usually be identifiable. The reason is that the node attributes $C$ are unobserved; e.g., we don't observe city of residency, and we cannot perfectly reconstruct it from the network structure. To circumvent this, we instead define the version of this estimand that we can (hope to) identify from the graph alone:

$$\psi_{t^*} := \frac{1}{n} \sum_i \mathbb{E}[Y_i \mid \mathrm{do}(T = t^*), G_n]. \tag{3.2}$$

This new estimand can be understood as taking the estimand in (3.1) and marginalizing out the information about $C$ that cannot be inferred from $G_n$. It is the average outcome under the hypothetical treatment, applied to the *same link structure* and to *a set of people consistent with the link structure*. In other words, it represents the peer contagion effect on the same graph with node properties only fixed to be consistent with the graph structure. Although this is somewhat less natural, we will see that it makes identification plausible.

There is an apparent significant drawback of the formalizations presented in Equations (3.1) and (3.2): both estimands are fundamentally tied to the particular sample available in our study, since both involve conditioning on the observed network $G_n$. We may be nervous that the formal quantity is tied to idiosyncracies of the particular observed network. In Theorem 1, we show that kind of sample dependency is not a serious issue. The approach is to show that both the causal estimands $\psi_{t^*}^{\text{full info}}$ and $\psi_{t^*}$ introduced in Equations (3.1) and (3.2) converge in probability to a fixed quantity as the size of the network $G_n$ increases. See Appendix A for the proof.

**Theorem 1** (Generalizing the sample-based causal estimands). *Consider an observed social network $G_n$, and let $Y_i$ be the labels associated with each node $i$. Assume the following*

    *(i) There exists $M > 0$ such that $\mathrm{Var}(Y_i) \leq M$, for all $i$;*

    *(ii) For any pair $(i,j)$ of nodes selected uniformly at random, $\mathbb{P}(\{\,i \text{ and } j \text{ share a neighbor}\}) \to 0$, as $n \to \infty$.*

*Then, there exists a fixed real number $\psi$ such that $\psi_{t^*}^{full\ info} \to \psi$ and $\psi_{t^*} \to \psi$, in probability.*

This theorem establishes that, provided the graph is sparse in a suitable sense, then both causal estimands converge to the same fixed quantity which does not depend on the particular network $G_n$. This decouples the interpretation of the causal estimand from the particular network under consideration. Additionally, it shows that the distinction between $\psi_{t^*}^{full\ info}$ and $\psi_{t^*}$ is not so important for practical applications—for large networks, the two values will anyway be very close. Note that this result is about proving that the *estimand* is well-defined, not about establishing convergence of some estimator.

## 4 Causal inference of peer effects using node embedding methods

Having formalized and motivated the target causal effect of interest, we now turn to sufficient conditions for causally identifying it.

As previously discussed, if we could exactly reconstruct the latent $C_i$ then identification would be straightforward. However, usually, this reconstruction is impossible. Happily, it is not necessary: an imprecise proxy for $C_i$ might still suffice for causal identification. Our goal now is to find sufficient conditions for an inferred proxy to enable identification. The next theorem gives such a condition. Informally, the condition is that we need a proxy $\lambda_i$ that carries enough information about $C_i$ so that the graph itself carries no further information about $Y_i$ after conditioning on $\lambda_i$. As we shall discuss in Section 5, node embedding methods can be viewed as a tool for constructing such $\lambda_i$.

We first define the following object, which plays a substantial role in the remainder of the paper.

**Definition 1.** *Let $V_i := S_Y(\{T_j : A_{ij} = 1\})^2$ be the aggregated treatment at node $i$, and $v_i^*$ its value under the hypothetical treatment intervention $T = t^*$. For a graph $G_n$, the* vertex conditional outcome model *is the function $m$ given by $m_{G_n}(v_i^*, \lambda_i) := \mathbb{E}[Y_i \mid v_i^*, \lambda_i]$.*

**Theorem 2** (Sufficient conditions for identification). *Let $G_n$ be the network, $A$ its adjacency matrix, and $Y_i$ the node-associated labels. Suppose that for each node $i$ we have a proxy for the latent attributes, $\lambda_i \in \mathbb{R}^k$, such that*

    *i  $Y_i \perp\!\!\!\perp A_{ij} | (\lambda_i, v_i^*)$, for all $i$ and $j$.*
    *ii  $P(V_i = v^* \mid \lambda_i) > 0$ for all $v^*$;*
    *iii  $\lambda_i$ is $C_i$-measurable.*

*Then $\mathbb{E}[Y_i|\mathrm{do}(T = t^*), G_n, \lambda_i] = m_{G_n}(v_i^*, \lambda_i)$.*

*Proof.* We sketch the proof and provide the full mathematical justification in Appendix B. Consider Figure 1 illustrating the core identification issue. The node-level causal effect $\mathbb{E}[Y_i|\mathrm{do}(T = t^*), G_n, \lambda_i]$ relies on conditioning on the network structure, and, in particular, on the edges $A_{ij}$. Doing so opens a backdoor path between $Y_i$ and $T_j$ (hence, also $V_i$) going through the unobserved $C_i$. The main idea of the theorem is that conditioning on $\lambda_i$ removes the effect of conditioning on the $A_{ij}$, and thus prevents the backdoor path from being opened. $\qquad\qquad\qquad\square$

In Theorem 2, condition (i) gives a notion of "sufficient amount of information about $C$" for causal adjustment, and is the main point of the theorem. Condition (ii) is the standard positivity assumption required for causal identification. Finally, condition (iii) is the condition that $\lambda_i$ is a partial reconstruction

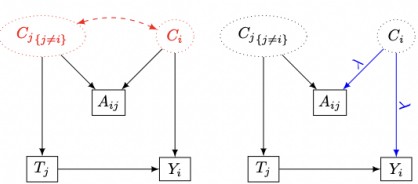

**Figure 1:** Identification of causal peer effects using embeddings.

---

²In general, the function $S_Y$ is set by the analyst for practical estimation purposes; this paper assumes $S_Y$ is the mean function.

of the node-specific attributes $C_i$. This condition is necessary to ensure $\lambda_i$ is not itself a collider for the latent node attributes.

This result readily extends to a condition for the identification of the peer contagion estimand $\psi_{t^*} = \frac{1}{n} \sum_{i=1}^{n} \mathbb{E}[Y_i | \mathrm{do}(T = t^*), G_n]$ introduced in Section 3. This is the content of the next result; see Appendix C for a proof.

**Corollary 3** (Identification of the causal estimand $\psi_{t^*}$). *Consider the same set-up and notation as that of Theorem 2. Then, the causal estimand $\psi_{t^*}$ can be identified as follows*

$$\psi_{t^*} = \frac{1}{n} \sum_{i=1}^{n} \mathbb{E}[m_{G_n}(v_i^*, \lambda_i)|G_n]. \tag{4.1}$$

Corollary 3 gives, in principle, a strategy for identifying the causal effect using the observed $\lambda$. Namely, learn $m_{G_n}(v_i^*, \lambda_i)$ and $\mathrm{P}(\lambda_i \mid G_n)$ and plug in to (4.1). This result is exact, and applies to any network. However, learning $\mathrm{P}(\lambda_i \mid G_n)$ seems difficult—it's unclear how to model this distribution. To circumvent the need to learn the unit-wise uncertainty about $\lambda_i$, we now give another result that again relies on the validity of the law of large numbers derived in Theorem 1. Intuitively, when the law of large numbers holds, we can replace averaging over the per-node uncertainty by averaging over all of the nodes (that is, the mean converges to the unit-level expectation).

We now state the main identification result as Theorem 4 below which is derived from both Theorem 2 and Theorem 1. We include its proof in Appendix D.

**Theorem 4.** *Consider the same set-up as that of Theorem 2. Furthermore, assume that the two conditions of Theorem 1 are also satisfied. Then $\lim_{n\to\infty} \frac{1}{n} \sum_{i=1}^{n} m_{G_n}(v_i^*, \lambda_i) = \psi$, in probability, where $\psi$ is the fixed real number such that $\lim_{n\to\infty} \psi_{t^*}^{full\ info}$ and $\lim_{n\to\infty} \psi_{t^*} = \psi$, in probability.*

## 5 Estimation of causal peer influence effects

We have now established a formal causal estimand $\psi_{t^*}$, and conditions on node-attribute proxies that enable identification. The basis for our estimation procedure will be Theorem 4, which says that—for suitable $\lambda$ and sufficiently sparse $G_n$—the statistical estimand

$$\widetilde{\psi}_{t^*} := \frac{1}{n} \sum_{i=1}^{n} m_{G_n}(v_i^*, \lambda_i) \tag{5.1}$$

approximates the causal estimand $\psi_{t^*}$, with the approximation becoming exact as $n \to \infty$. Recall $m_{G_n}(v_i^*, \lambda_i) := \mathbb{E}[Y_i \mid v_i^*, \lambda_i]$. Our task is to estimate $\widetilde{\psi}_{t^*}$ from the observed data using the observed treatments and outcome $(T_i, Y_i)_{i=1:n}$ and the link structure $G_n$.

To achieve this, we use embedding-based semi-supervised prediction models to learn $\hat{\lambda} \in \mathbb{R}^k$ and $\hat{m}_{G_n}(v^*, \hat{\lambda}_i) \approx \mathbb{E}[Y_i \mid v_i^*, \hat{\lambda}_i]$. Then, we plug these estimates into (5.1) as $m$ and $\lambda$. For concreteness, we describe a particular approach based on Veitch et al. [Vei+19] and Veitch et al. [VWB19]. The estimation procedure follows three main steps:

**Step 1.** We train a model using relational empirical risk minimization [Vei+19] to jointly learn the embeddings $\hat{\lambda}_i$ and $\hat{m}_{G_n}(v_i, \hat{\lambda}_i)$. To do this, we first compute $v_i = S_Y(\{t_j : A_{ij} = 1\})$ at each vertex $i$—i.e., we create a new aggregate treatment feature and attach it to each vertex.

Relational ERM works by minimizing a risk defined as an expectation over randomly sampled subgraphs of the larger network. Let $\mathtt{Sample}(G_n, k)$ be a sampling algorithm that returns a random subgraph of size $k$ from $G_n$ (e.g., the subgraph induced by a random walk of length $k$). Let $v_{G_k}$ be the set of vertices of the subgraph $G_k$. We then define the loss function on the $G_k$ to be

$$L(G_k, \lambda, \gamma) = q \cdot \sum_{i \in v_{G_k}} (y_i - m(v_i, \lambda_i; \gamma))^2 + \sum_{i,j \in v_{G_k} \times v_{G_k}} \mathrm{CrossEntropy}(A_{ij}, \sigma(\lambda_i^T \lambda_j)), \tag{5.2}$$

where $q \in [0, 1]$ and $\sigma$ is the sigmoid function. The second term is a network reconstruction term that extracts node-level information from the network by requiring the embeddings to be predictive

of the edge structure. The first term learns a predictor $m$, parameterized by $\gamma$ (global parameters shared across the network), that predicts $y_i$ from the aggregated treatment $v_i$ and the embedding $\lambda_i$. In our experiments, $m$ will be a linear predictor, and $\gamma$ the (parameters of the) linear map. When $y$ is binary, the binary cross-entropy loss is used instead of the MSE loss. Although the map from the embeddings is linear, because the embeddings are unconstrained, the combined embedding-linear map can flexibly represent complex relationships between the network structure and outcome/treatment. In Section 6 we see that the model ably captures a relationship between $Y$ and a latent attribute $C$ that is non-linear in $C$, where the relationship between the network and $C$ is complex.

Then, we train the model by fitting

$$\hat{\lambda}, \hat{\gamma} = \operatorname*{argmin}_{\lambda, \gamma} \mathbb{E}_{G_k = \texttt{Sample}(G_n, k)}[L(G_k, \lambda, \gamma) | G_n]. \tag{5.3}$$

**Step 2.** We define $\hat{m}_{G_n}(v_i^*, \lambda_i) = m(v_i^*, \lambda_i; \hat{\gamma})$, where $v_i^*$ is the value of $v_i$ attained under the interventional treatment $T = t^*$. Then, for each unit vertex $i$ in $G_n$ we compute $\hat{m}_{G_n}(v_i^*, \hat{\lambda}_i)$.

**Step 3.** Finally, the peer influence estimate is the average over the estimated vertex conditional outcomes $\hat{m}_{G_n}(v_i^*, \hat{\lambda}_i)$:

$$\hat{\psi}_{t^*}(G_n) = \frac{1}{n} \sum_i \hat{m}_{G_n}(v_i^*, \hat{\lambda}_i).$$

## 5.1 Validity of the estimation procedure

In this section, we informally discuss the validity of the estimation procedure described in the previous section. More specifically, recall that, in Theorem 1, we showed that both causal peer influence estimands proposed, $\psi_{t^*}^{\text{full info}}$ and $\psi_{t^*}$, converge, in probability, to a fixed real number $\psi$, as $n \to \infty$. The question that remains is whether the estimator proposed, $\hat{\psi}_{t^*} = 1/n \sum_{i=1}^{n} \hat{m}_{G_n}(v_i^*, \hat{\lambda}_i)$, also converges to $\psi$, in probability, as $n \to \infty$. This will clearly hold as long as $\hat{m}_{G_n}(v_i^*, \hat{\lambda}_i) \to m_{G_n}(v_i^*, \lambda_i)$ for some $\lambda$ satisfying the conditions of Theorem 2.

Why should one expect the above consistency property? First, consider $\hat{m}(v, \lambda) \to \mathbb{E}[Y_i \mid v, \lambda]$. The map $m$ is defined as the minimizer of the squared error empirical risk. The minimizer of the true risk is the required conditional expectation. Accordingly, this part of the learning is, essentially, the same as what occurs with non-network data and is plausible.

The trickier judgement is why can one expect that $\hat{\lambda}_i \approx \lambda_i$ for some $\lambda$ satisfying the conditions of Theorem 2? That is, is it plausible that the learned embeddings behave as proxies for the node-level attributes (i.e., are $C$-measurable), and carry sufficient information to adjust for the homophily effect (i.e., $Y \perp\!\!\!\perp A_{ij} \mid (\lambda_i, v_i^*)$)?

The $C_i$-measurability is just the requirement that each node-level embedding captures unobserved information specific to that node. This is exactly the aim of embedding methods, to leverage network structure in order to capture the latent properties of each node [HYL17]. Accordingly, it is reasonable to assume that the embedding-based method described in Section 5 satisfies this property (at least, asymptotically). With respect to whether the embeddings carry sufficient information to adjust for the homophily effect when estimating peer influence, we note the following: in order to satisfy this condition, it suffices that the embeddings are predictive of the outcome or of the edge structure. To see why this is plausible, recall the expression of (5.3) used to fit the embeddings. The cross-entropy term encourages the embeddings to be sufficient (at the graph level) for predicting the edge structure. The squared error term encourages the embeddings to be sufficient for predicting the labels $Y_i$. Hence, the objective function is aimed at exactly the requisite information.

Unfortunately, despite the intuitive justification above, there is no firm guarantee that any given embedding method actually achieves its conceptual goal. In effect, by working with embedding methods one is trading off the precise guarantees enabled by parametric models (exploited in [SM16]) in favor of techniques that have good empirical performance even in the (typical) case that the networks have structure that is inconsistent with parametric specification.

In fact, for the specific case of relational ERM we can say something further. Davison and Austern [DA21] study the asymptotics of the embedding learning procedure under the assumption that $G_n$ is

**Table 1:** The embedding-based estimator $\hat{\psi}_{t*}$ effectively adjusts for confounding and recovers the true treatment effect. The ground truth value of peer contagion is 1. Zero, low, and high confounding levels correspond to $\beta_1 = 0, 1$, and $10$, respectively. For $\hat{\psi}_{t*}$, the reported values represent the mean over 100 different global random seeds. The seed for the simulated treatment and outcome data is kept constant. For the Unadjusted and Parametric estimators, the reported values represent the respective estimated regression coefficients for the aggregated treatment used when predicting $Y$.

| | district | | | age | | | join_date | | |
|---|---|---|---|---|---|---|---|---|---|
| Conf. level | **Zero** | **Low** | **High** | **Zero** | **Low** | **High** | **Zero** | **Low** | **High** |
| Unadjusted | 0.99 | 1.64 | 7.40 | 1.00 | 1.39 | 4.90 | 0.99 | 1.38 | 4.81 |
| Parametric | 0.99 | 1.41 | 5.28 | 1.00 | 1.33 | 4.20 | 0.98 | 1.28 | 4.00 |
| $\hat{\psi}_{t*}$ | 0.84 | 0.96 | 1.17 | 0.94 | 0.94 | 1.11 | 1.01 | 1.03 | 1.10 |

generated by an exchangeable random graph, or a graphon (a broad family of models that includes, e.g., stochastic block models and latent space models [Bor+16; CCB16; CF17; CD18; Jan18]). Informally, Davison and Austern [DA21] show that, asymptotically, the embeddings are functions of node-specific attributes $C_i$ alone and that these embeddings capture the limiting distribution of the network (see Davison and Austern [DA21] Thm. 7). Accordingly, in this case, the learned embeddings satisfy the conditions of Theorem 2.

## 6 Experiments

To study the practical performance of the estimation procedure described in Section 5, we conduct experiments using semi-synthetic data. We use a subset of data from the Slovakian social media website *Pokec* [TZ12; LK14]. The data includes both links between users and limited node-level covariates. Our basic strategy will be to use these node level covariates as the latent $C_i$'s. To do this, we simulate treatment and outcomes for each unit in a manner that depends on $C_i$. At estimation time, we hide the $C_i$ variables, and check how well we can recover the true causal effect using only the graph data and observed treatments and outcomes.

We extend the setup of Veitch et al. [VWB19] who applied a similar relational empirical risk minimization based technique in order to adjust for the network unobserved confounding effect when performing average treatment effect (ATE) estimation of $T_i$ on $Y_i$. That work assumes pure homophily, whereas this paper examines peer-based contagion effect estimation.

Following Veitch et al. [VWB19] we analyze a sub-network of approximately 70000 users connected by roughly 1.3 million links, representing a subset of the original Pokec data restricted to the districts Žilina, Cadca, and Namestovo, from the same region. The analysis consists of two parts:

1. estimating the peer contagion effect of binary treatments on continuous response variables, varying both the source of confounding and its strength;
2. estimating the peer contagion effect of binary treatments assigned at time $t_1$, the beginning of a study period, on treatments assigned at a subsequent time $t_2$, the end of that study period.

See supplement (Appendix F) for additional experimental details.

### 6.1 Continuous outcome

For the first set of experiments, the treatment $T$ and outcome $Y$ values are generated from three variables (one at a time), taken as the unobserved confounders $C$: district, age, and Pokec join date. For each of these three confounders, we standardize, then bin each of them into three possible values: $-1, 0$, and $1$. We then apply a function $g$ which transforms them into probabilities: $g(C) = 0.5 + 0.35 \cdot C$, so that $g(C) \in \{0.15, 0.5, 0.85\}$. Then, we simulate according to:

$$
\begin{aligned}
T_i &= \text{Bern}(g(C_i)); \\
V_i &= \text{Average}(T_j : A_{ij} = 1); \\
Y_i &= \beta_0 \cdot V_i + \beta_1 \cdot g(C_i) + \varepsilon_i, \ \varepsilon_i \sim N(0, 1).
\end{aligned}
\tag{6.1}
$$

The parameter $\beta_0$ controls the amount of peer influence, while $\beta_1$ measures the amount of network confounding. In this simulation setup, the causal peer contagion effect of interest, $\hat{\psi}_{t*}$ is precisely equal to $\beta_0$. We fix $\beta_0 = 1$ as the *ground truth* value of confounding, and let $\beta_1$ vary in $\{0, 1, 10\}$ corresponding to no, low, and high contagion, respectively.

Since we only observe $T_i$, $V_i$ and $Y_i$ we test whether $\hat{\psi}_{t*}$, the estimator proposed in Section 5, adjusts for the unobserved confounding due to $C$, and accurately estimates $\beta_0 = 1$. For each combination of $C$, and strength level $\beta_1$, we obtain simulated $V_i$ and $Y_i$ values, then we train the embedding-based relational empirical risk minimization model described in Section 5, using a random walk sampler with negative sampling, and the default settings of Veitch et al. [VWB19]. Finally, we use the trained model to predict two sets of adjusted $Y$ values:

1. $Y_i$'s resulting from the aggregated treatment $v_i^*$, when all $T_i$ are set to 0, i.e. $\hat{m}(\hat{\lambda}_i, v_{\{i,t_i^*=0\}}^*)$;

2. $Y_i$'s resulting from the aggregated treatment $v_i^*$, when all $T_i$ are set to 1, i.e. $\hat{m}(\hat{\lambda}_i, v_{\{i,t_i^*=1\}}^*)$.

The causal effect estimator of interest is then $1/n \sum_{i=1}^n \hat{m}(\hat{\lambda}_i, v_{\{i,t_i^*=1\}}^*) - 1/n \sum_{i=1}^n \hat{m}(\hat{\lambda}_i, v_{\{i,t_i^*=0\}})$.

We compare the results of this procedure against those of two baseline estimators. The first one is the naive estimator which does not adjust for confounding, namely $1/n \sum_i \mathbb{E}[Y_i | v_{\{i,t_i^*=1\}}^*] - 1/n \sum_i \mathbb{E}[Y_i | v_{\{i,t_i^*=0\}}^*]$, where expectations are the estimated values obtained via linear regression. The second baseline seeks to mimic the parametric peer contagion estimation procedure of Shalizi and McFowland III [SM16], by fitting a mixed-membership stochastic block model to the Pokec data, inferring 128 latent communities (128 is equal to the dimension of the embeddings), then predicting $Y$ via a linear regression on the aggregated treatment and inferred latent community values.

Table 1 summarizes the results of the embedding method compared to those of the above baselines. Whenever confounding was present, $\hat{\psi}_{t*}$ yielded the most accurate results out of all three estimators, by a significant margin. This confirms that the nonparametric method can leverage the network structure and obtain reliable peer contagion estimates. With no confounding present, all three estimators produced valid results, yet, when district and age were used as unobserved confounders, $\hat{\psi}_{t*}$ was slightly less accurate compared to the baselines. Indeed, if there is no unobserved confounding, then no network-based adjustment is necessary; the variation due to embedding-model fitting causes errors that simpler methods do not suffer from. In Appendix E, Table 3, we provide error bars for the average estimated peer effects obtained in Table 1, which aim to illustrate the consistency of our results. The errors reported for $\hat{\psi}_{t*}$ represent the standard errors computed over 100 different global random seeds, while the seed for the simulated treatment and outcome data is kept constant. For the Unadjusted and Parametric estimators, the reported error bars represent the standard errors of the estimated regression coefficients for the aggregated treatment used when predicting $Y$. The main caveat is that due to the relational, non-i.i.d. structure of our data, these errors do not represent proper confidence bands, and should be interpreted with caution. To obtain valid confidence intervals, proper asymptotic normality results for the studied estimators are necessary.

## 6.2 Binary outcome: a model for peer influence and vaccination

Recall the motivating example in Section 1: some proportion of the observed population gets vaccinated at time $t_1$, while others get vaccinated at a subsequent time $t_2$. We now illustrate how to use node embeddings to study the peer influence of the former group on the latter. We use the following data setup:

1. Simulate the treatment by the same procedure as in Section 6.1, i.e. as a function of the unobserved confounders;
2. Randomly select 50% of the vertices, and set $Y_i = T_i$, but then delete $T_i$ for those respective nodes. The uncensored treatment values $T_i$ correspond to individuals who were vaccinated at $t_1$, while the $Y_i$ values represent individuals vaccinated at $t_2$;
3. Compute the causal effect of $V_i$ on $Y_i$ only for the vertices where $Y_i$ is defined.

The above design matches the following scenario: whether or not an individual gets vaccinated depends on latent attributes such as their age, or location. Since $Y_i = T_i$, there is no causal peer

**Table 2:** The embedding-based estimator $\hat{\psi}_{t*}$ accurately recovers the true peer contagion effect of binary treatments on other subsequent binary treatments. The ground truth peer contagion value is 0. For $\hat{\psi}_{t*}$, the reported values represent the mean over 100 different global random seeds. For the Unadjusted and Parametric estimators, the reported values represent the respective estimated regression coefficients for the aggregated treatment used when predicting $Y$.

| Peer influence on vaccination | district | age | join_date |
|---|---|---|---|
| Unadjusted | 2.03 | 0.12 | 0.68 |
| Parametric | 1.30 | 1.03 | 0.98 |
| $\hat{\psi}_{t*}$ | 0.09 | 0.11 | 0.22 |

influence effect of the neighbors $j$ on node $i$. Despite the lack of a causal relationship, an association is clearly present due to the dependence on latent confounders. The question is whether the proposed method can adjust for the unobserved causes and render the true causal effect.

Using the same baseline estimators, and an analysis similar to that in Section 6.1, we obtain estimates for the peer contagion effect, summarized in Table 2. We note that, for all three hidden confounders, the embedding-based method produces the estimates closest to the ground truth value of 0. For the "age" covariate, the naive, unadjusted estimate is very close to the embedding-based one, showing that age may not be the optimal predictor for network tie formation, preventing the non-parametric method from appropriately adjusting for the network structure, and yielding our method similar to the naive one, in this situation. We also note that, when registration date is used as a hidden confounder, all three estimates are less accurate, suggesting that "join_date" may also not be the best predictor of the network edges. Nonetheless, in this scenario, the nonparametric estimator still has superior performance over the two baselines. In Appendix E, Table 4, we provide error bars for the variation induced by the choice of random seed, obtained in a similar way to those corresponding to the continuous outcome scenario.

In Appendix G we perform additional analyses further varying the values of $\beta_1$ and noise level $\varepsilon$, as well as illustrate the performance of our method on the Wikipedia hyperlink network [Yin+17].

## 7 Related work

There is an active literature on estimating causal effects from networked data [ZA21].

**Randomized controlled trials.** To estimate causal effects under interference and network confounding, several recent works rely on designing randomized experiments. Toulis and Kao [TK13] use potential outcomes and a sequential randomization design to estimate peer influence on observed networks. Eckles et al. [EKB16] develop an experimental design to study the impact of peer feedback on the behavior of Facebook users. Fatemi and Zheleva [FZ20b] propose an experimental design for estimating the effect of the treatment alone on the unit's outcome, isolating it from peer effects, while Fatemi and Zheleva [FZ20a] discuss an approach of minimizing selection bias when performing A/B testing on networks.

**Causal network inference from observational data.** Within observational settings, van der Laan [van14], Ogburn et al. [Ogb+17], Tchetgen Tchetgen et al. [TFS17], and Tran and Zheleva [TZ22] tackle the issue of causally estimating social contagion from observed networked data, yet they assume there are no latent sources of network confounding which affect both the treatment and the outcome, by considering all node-related features as known. This is a significant limitation which this paper seeks to address.

Other works which account for latent confounding assume that the network (and, implicitly, the unobserved confounders) can be represented using parametric methods. Shalizi and McFowland III [SM16] and Shalizi and Thomas [ST11] use stochastic block models and latent space models in order to obtain consistent estimators of peer contagion. Building upon this work, Sridhar et al. [SBB22] adjust for the unobserved confounders in peer influence estimation by using Poisson Influence Factorization (PIF) models.

Some of the recent studies which address nonparametric peer influence estimation from observational data propose different approaches which complement the node embedding based method proposed in this paper. Eckles and Bakshy [EB17] estimate peer effects on hundreds of millions of observations from Facebook data by using high-dimensional adjustments for covariates via propensity score models. One limitation of this work is that covariates are readily available, whereas this paper seeks to cover the scenario in which they are unobserved. Egami and Tchetgen Tchetgen [ETT21], on the other hand, tackle the situation of unobserved confounding, by using negative control outcomes and exposure variables to estimate contagion effects. This approach complements the methods of this paper. While not directly related to peer influence estimation, Guo et al. [GLL20] also propose a nonparametric technique of adjustment for the unobserved features that affect network treatment and outcome. They propose using "graph attentional layers" in order to map the network information to a partial representation of the hidden confounders and use it to perform counterfactual evaluation.

## 8   Discussion

This paper studies the problem of causally estimating peer influence from observational data in the presence of unobserved confounding. The main contributions of this work are

1. formalizing and justifying two causal estimands for contagion while accounting for homophily (Section 3);
2. giving sufficient conditions for network embeddings to enable causal identification (Theorem 2);
3. illustrating how embeddings can be used to estimate peer contagion and showing their practical performance is overall better than that of standard naive and parametric estimation techniques (Section 5 and Section 6).

There are several avenues for further work. First, one current methodological limitation is the lack of an asymptotic normality result for the peer influence estimator $\hat{\psi}_{t^*}$ defined in Section 5. Characterizing the asymptotic normality of the main estimator would allow one to construct valid confidence intervals around the estimated values, at a desired significance level. Secondly, this paper only focused on network embeddings learned via subsampling as a proposed nonparametric method for peer contagion estimation. An important future work direction is exploring other nonparametric techniques (e.g. graph neural networks [Zho+18; Wu+19; ZCZ22; Ma+22; Hus+21]) which leverage the homophily present in social networks and are able to learn meaningful node representations to be used downstream for peer influence estimation.

## 9   Acknowledgements

This work was partially supported by Open Philanthropy. We thank Steve Yadlowsky for helpful comments and feedback. We would also like to thank the Research Computing Center (RCC) at The University of Chicago for computing resources.

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
