# OpenReview forum: "Using Embeddings for Causal Estimation of Peer Influence in Social Networks"
_NeurIPS.cc/2022/Conference — NeurIPS 2022 Accept_

### Official Review · Reviewer_9be6 · 2022-07-06

**Rating:** 7
**Confidence:** 3
**Soundness:** 3 good
**Presentation:** 4 excellent
**Contribution:** 2 fair

**Summary:**

This paper presents a formulation for casual estimation of treatment effects in the presence of network influence. The proposed formulation allows for the covariates affecting both network formation and treatment adoption to be unseen and estimated from an observed network. More specifically, the proposed method uses network embeddings as a means of accounting for network influence in a causality model.

**Questions:**

In addition to the points raised in the weaknesses, one other question may be helpful for future work. Can the proposed method be extended to the case where there is more than one network observed, maybe with some assumptions on independence between the networks? For scenarios like social media, there are often several networks present (following, friendship, replying, etc.) all of which may have confounding effects on treatment.

**Limitations:**

See the strength and weaknesses section about limitations for use in practical settings. The authors have addressed any limitations with regard to potential negative societal impacts.

**Strengths And Weaknesses:**

___Strengths___: The paper presents a clear, relatively novel method that is well supported theoretically. The use of network embeddings is a straightforward extension of incorporating network structure into a causality model based upon previous works that used techniques like stochastic blockmodels (SBMs) or latent space models to incorporate network information. The proposed method, in contrast to previous ones, does not require as many assumptions about the nature of the network being incorporated into the model as other techniques, like SBMs do. The paper is also attacking an important problem, that of determining causality in treatments in network settings, which has important implications for very serious problems like understanding vaccine choices.

The paper also has strong theoretical support for the proposed method. The convergence of the various forms of the estimand only requires mild, often true assumptions like the relatively large size of the network and sparsity of the network. Also, the incorporation of a loss function into the ERM embedding that supports the causal inference model is also noteworthy and makes the method more complete.

Finally, the article's presentation is clear and the writing is generally free of errors.

___Weaknesses___: The paper does not present a thorough empirical validation and has some practical limitations that are not addressed. The paper only considers one empirical data set and artificially crafts the treatment based on observed covariates. A more convincing empirical validation would feature more than one data set and, ideally, one that would have observed treatment effects from a real-world scenario, like vaccine uptake or voicing of political support of a candidate, or something else similar. Additionally, the paper does not investigate the nature of the embedding procedure on its results; the paper only considers one type of deep learning-based embedding technique. Thus it's not clear how much the embedding technique can affect the results.

The paper also does not address some important real-world limitations of the proposed model. For one, not all node attributes contribute to link formation in any given network. For example, the attribute of gender is usually inversely related to link formation in something like a sexual contact network. Additionally, this problem can be further compounded by non-assortativity in networks, which is another occurrence in real-world social networks. Furthermore, within the empirical results the paper identifies that when there is zero confounding, the performance of the proposed method suffers. So, it would seem there is a significant limitation in this method’s usage that can have deleterious effects on results in actual usage.

---

> ### Author Response · Authors · 2022-08-02
> **Rebuttal**
>
> Thank you for the review and questions! We’re glad you found the paper clear and that you agree that the topic is important.
>
> In our view, the main contribution of the paper is giving a formalization and identification argument for contagion effects that does not rely on a parametric network model. This is not a “straightforward extension” of the parametric results! In particular, earlier work either relied on the parametric model to even define what the causal effect is, or treated the graph as non-random (thus not accounting for homophily). Formalizing both the effect and the identification is quite subtle because it requires the estimand itself to be a (graph-measurable) random variable. This is the main novelty and contribution of the paper. In particular, the method we propose is meant to demonstrate the power of this formalization. It is certainly possible that our formalization and identification techniques could be used as the basis for some other more sophisticated, better, estimation algorithm---this would be an interesting direction for future work!
>
> In this context, we think that our experimental evaluation is appropriate.
>
> Notice that the dataset is only semi-synthetic. The relationship between the node attributes---region, age, join date---and the network is fixed by the real-world process that generated the graph. The simulation is based only on the latent attribute, and does not bake in any favorable structure relating the treatment and outcome assignment and network structure.
>
> Also notice that the relationship between the network and each of these attributes is presumably quite different---we do not rely on particular simple network/latent structures. Whether a latent attribute causes associative, or anti-associative behavior is not a priori relevant. All that matters is the ability of node embedding methods to extract this information from the network. Moreover, a key point of the paper is that if there is some latent information that is not associated with edge formation then this latent information will also not be confounding. This is a key idea that allows us to use network-constructed proxies in place of the unobserved latent variables.
>
> We have also added ablation experiments in the appendix varying the noise and $\beta_1$ parameters, per the suggestion of reviewer 6sfM. These suggest that the embedding method accurately recovers the true contagion effect and performs better than baselines even for small (but non-zero) confounding levels, and also for a range of low to high signal to noise ratios (see Tables 5 and 6 in Appendix G1 of the Appendix).
>
> Furthermore, we have also tested our method on a new dataset, the Wikipedia hyperlink network dataset (see Appendix G2 of the Appendix). Even for a new dataset, with a different network structure than the Pokec data, the method accurately estimates the ground truth peer contagion effect. This confirms the wider applicability of the proposed technique.
>
> * "In addition to the points raised in the weaknesses, one other question may be helpful for future work. Can the proposed method be extended to the case where there is more than one network observed, maybe with some assumptions on independence between the networks? For scenarios like social media, there are often several networks present (following, friendship, replying, etc.) all of which may have confounding effects on treatment."
>
> The key question here is which network the contagion effects are assumed to carry over. Given any answer to this question, it should be straightforward to adapt the multi-link structure by using an embedding method that captures multi-link behavior. This would be an interesting direction for future work!

---

> > ### Author Response · Authors · 2022-08-08
> > **Questions about rebuttal**
> >
> > Thank you again for your review. Do you have any further questions?

---

> > > ### Comment · Reviewer_9be6 · 2022-08-08
> > > **Further Questions**
> > >
> > > After re-reading back through Section 2, and especially the equations in 2.1, I believe I see the point of "Whether a latent attribute causes associative, or anti-associative behavior is not a priori relevant." How $C$ affects the link formation in $A$ could really be essentially anything (homophilic, assortative, non-assortative, etc.). That said, the second sentence in the rebuttal highlights a really important assumption about the method in practice: "All that matters is the ability of node embedding methods to extract this information from the network." Knowing that the embedding is capturing the right aspects of the network, relative to the confounding variable, doesn't seem to be trivial knowledge. So, what happens in the scenario when the network does not form links in a homophilic manner, relative to the observed variable, but is maybe homophilic to a different variable that works opposite to the one observed?

---

> > > > ### Author Response · Authors · 2022-08-08
> > > > **Response to further questions**
> > > >
> > > > As a practical matter for the choice of embedding method, a key consideration is the ability of the embedding method to capture the relationship between latent network-associated variables and the outcome variables. In general, there's not much known about this theoretically, though see [Davison and Austern 2022 - Asymptotics of Network Embeddings Learned via Subsampling] for an argument that embedding methods work well with non-parametric models (graphons), so we do not need to rely on particular homophily assumptions. Additionally, a key observation of the paper is that (information in) node-related latent variables that do not influence edge formation are not confounding, so it's not necessary to reconstruct this information (see Thm. 2).
> > > >
> > > > To be clear, the only observed variables used in the inferential procedure are the outcome, treatment, and network structure---$C$ is never observed. In the experiments presented, neither the age nor gender variables are particularly homophilic (e.g., running a clustering can detect region, but not other node attributes). The Pokec experiments run exactly the same code on exactly the same network; the results show that the fact that the network is strongly homophilic with respect to region does not interfere with good estimates when the true confounding variable was age or sex.
> > > >
> > > > More fundamentally though, the main point in this paper is the non-paramatric formalization and identification of contagion effects. It may be possible to construct network structures where a different embedding model is required. However, even if that was the case, a new, alternative, embedding method could be used in combination with the presented formalization and identification arguments to estimate contagion. Generally, causal estimation requires formalizing an effect, providing an identification argument, and finding an effective estimation strategy. The main contribution of this paper is formalization and identification. We view the estimation strategy as mainly "proof of concept". Though it is remarkable that, in practice, even this simplest implementation works well!

---

> > > > > ### Comment · Reviewer_9be6 · 2022-08-09
> > > > > **Concerns Addressed**
> > > > >
> > > > > I believe my primary concern is addressed. I am updating my review accordingly.

---

### Official Review · Reviewer_4Cyd · 2022-07-11

**Rating:** 7
**Confidence:** 3
**Soundness:** 3 good
**Presentation:** 4 excellent
**Contribution:** 3 good

**Summary:**

This paper present a method that use node embedding to causally estimate the peer influence in social networks. In this task, the key challenge is to distinguish the similarity of linked users and peer influence. This paper claims that for a graph that is sparse enough, the hidden attributes can be ignored if we can provide sufficient information on graph structure. And then under this theoretic results, they propose that jointly adding a term relugarizing the node embedding to help reconstruct the graph structure can help us distinguish the peer similarity and peer influence. Experimental results on a semi-synthetic show that the proposed method can bring a significant boost.

**Questions:**

How is the q in Eq. 5.2 decided if we do not have ground truth causal effect?

**Limitations:**

See the summary.

**Strengths And Weaknesses:**

Strengths:
1. The topic of this paper is interesting.
2. The presentation of this paper is good.
3. The idea of this paper is easy to understand, yet with theoretical support.

Weakness:
1. In this framework, how should we remove the bias brought by the correlation between treatment and covariates? In traditional frameworks, such a challenge is usually addressed by inverse probability of treatment weights (IPTW). But I do not think that computing IPTW in the proposed framework is trivial.
2. The experiments are conducted on a synthetic dataset with real-world graph structure. I am wondering if it is possible to run the model on a complete real-world dataset. I understand that on real-world dataset, it is hard to get ground-truth labels. However, we can try to use the model to identify some social effect that are found by social scientists.

---

> ### Author Response · Authors · 2022-08-02
> **Rebuttal**
>
> Thank you for your review! We’re happy you found the paper interesting, clear, and well supported.
>
> With respect to your questions:
>
> Removing the bias due to association of treatment and latent covariates is the main point of the paper, and is indeed pretty non-trivial! :) To handle observed node-level covariates, we only need to incorporate them into the $m_G$ function explicitly; e.g., with a neural network.
>
> Notice that the network dataset is only semi-synthetic. The relationship between the latent factor $C$ and the network is entirely determined by the real, physical, process that generated the network. The treatment and outcome are determined by this $C$, which is totally unknown to the model.
>
> Lastly, the parameter $q$ in Eq. 5.2 is set to 0.005 matching previous work. We clarify this in Appendix F.

---

### Official Review · Reviewer_6sfM · 2022-07-12

**Rating:** 5
**Confidence:** 3
**Soundness:** 3 good
**Presentation:** 3 good
**Contribution:** 3 good

**Summary:**

The authors tackle the well-known challenging problem in social networks that influence and homophily effects are not trivially distinguishable. In particular, this problem is more challenging because confounding factors are typically unobserved. The proposed method uses the embeddings to represent latent confounding factors and adjusts such confounding factors by adding a network regularization term in the estimation of causal influence effects. Authors show the validity of the proposed method by providing enough theoretical analysis. Also, empirical analysis under semi-synthetic experimental setting verifies the benefits of the proposed method as opposed to baseline estimations.

**Questions:**

In general, experimental part is the weakest part throughout the manuscript. Here are some suggestions regarding the experiments.

- While experimental results can be affected by network structure and the relationships between networks and node attributes, authors need to present richer experimental results on various datasets and simulation models. In particular, since contagion is simulated, the proposed experiment should be applicable for many public datasets.
- Also, variations on the parameter values of \beta's would be great to check. Currently, only \beta_0 = 0 or \beta_0 >= \beta_1 cases are present. However, from the indecisive results for "Zero" case, we may think that the proposed method might be better when \beta_0 is large enough compared to \beta_1 but not necessarily good enough for small \beta_0 case. Further experiments can check this out.
- Moreover, the fixed \beta_1 value does not show the performance over SNR (signal-to-noise ratio; here it should be \beta_0 over the standard deviation of the noise). It would be interesting to see whether SNR impacts the performance more or \beta_{1} / \beta_{0} impacts more.
- Isn't the formula in page 8 eventually \beta_{0}? If so, it would be great to explicitly describe so in order to prevent any confusion. Before that formula, everything is described with respect to \beta, but suddenly it seems as if a new estimator were introduced. Actually, I would suggested introducing this in the earlier part of the paper, instead of in the experiment section.

Here are some other questions.
- Formula (5.2) and experimental setup is based on the continuous variable regression. However, the introductory example is based on the binary outcome. Should Formula (5.2) remain the same regardless of the type of Y values? How about the simulation process in Page 7? Without some management, Y_{i} cannot be guaranteed to be binary.
- In Page 4, line 128, authors mention that the identification of causal effects would be straightforward if we know the latent C_i's. However, the identification does not seem trivial when we do not know the impact of each C_i's on the final response variable. Authors need to provide explanation on this argument.
- As described above, the choice of q value is not straightforward. Authors need to provide some guidance.
- Authors describe that

**Limitations:**

Authors somehow present the limitation of the proposed method through the discussion about experimental results. However, it would be great if authors can summarize the limitation of the proposed methods.

**Strengths And Weaknesses:**

Strengths
- Authors address a very well-known but critical problem, which fits well into the conference. The final estimation method is straightforward and easily understandable.
- Authors provide enough theoretical support when developing each step of the proposed method. Such support helps understanding without any big logical gap.
- Experimental results show that the proposed method outperforms baseline methods, particularly when the confounding factors become problematic in the estimation of peer influence effect.

Weaknesses
- While notations are clearly defined in theory, it is not easy to follow the meaning or condition (e.g. what kind of value that each variable can have) of  some variables. Some examples would be helpful for straightforward understanding.
- Experimental setup needs to be enforced because the current results show limited aspect of the proposed method. Please take a look at the suggestions below.
- Unlike the typical ML problem, the parameter q is not trivially tunable through AutoML or any hyperparameter tuning because q balances two non-comparable objectives. Authors need to provide some guidance.

---

> ### Author Response · Authors · 2022-08-02
> **Rebuttal**
>
> Thank you for your review. We’re glad you agree that the contagion estimation problem is a critical open problem, and that the final estimation method is straightforward and easily understandable.
>
> With respect to the experimental evaluation, we emphasize that the contribution of this paper is primarily showing how to formalize and causally identify the contagion effect in this setting without appealing to some parametric network model. The purpose of the particular method implementation is to show that this formalization can yield a simple and efficient procedure by adapting node-embedding methods. In particular, we are _not_ claiming that this is the best possible way of doing this adaptation.
>
> Note though that the experiments do provide strong evidence that the existing method can indeed handle rich structure. The relationships between the network and the various attributes we consider for the Pokec data (region, age, registration date) are quite varied---we are not relying on some special form relating attributes to network structure.
>
> In addition, following your suggestions, we have added supplementary analyses varying the confounding level $\beta_1$ and the noise level for the Pokec dataset (see supplementary material, Appendix G1), as well as a new experiment on a completely different dataset (see Appendix G2). For the ablation analyses in Appendix G1, we first investigate the performance of our method under small, but non-zero confounding levels $\beta_1$ in {0.25, 0.5, 0.75}. We find that the method consistently outperforms baselines in these cases, suggesting that slight underperformance in the $\beta_1 = 0$ case in the paper is an edge situation (see Table 5 in the Appendix). Furthermore, for a fixed confounding level and region, we vary the noise level in order to ensure a wide variety of signal to noise ratios (both high and low). We find that across all noise levels, the embedding method most accurately estimates the peer contagion effect compared to the baselines (see Table 6).
>
> Finally, we have also applied our method to a new dataset based on the Wikipedia hyperlink network, where each node is an article and an edge is a hyperlink between two articles. The hidden confounder is taken to be the unique category tagged to each article. Even for this new dataset, which presents a different graph structure than the Pokec data, the method accurately estimates the ground truth effect (see Table 7).
>
> In short: both the original and supplementary analysis provide strong evidence that the theoretical arguments in the paper can be readily translated into practical estimation procedures.
>
> * "In Page 4, line 128, authors mention that the identification of causal effects would be straightforward if we know the latent C_i's. However, the identification does not seem trivial when we do not know the impact of each C_i's on the final response variable. Authors need to provide explanation on this argument."
>
> In this case, the task would “just” be standard backdoor adjustment. Although this is still non-trivial as a statistical task, the identification argument is standard.
>
> * "Unlike the typical ML problem, the parameter q is not trivially tunable through AutoML or any hyperparameter tuning because q balances two non-comparable objectives. Authors need to provide some guidance."
>
> This is an excellent point. In all of our experiments, we fix $q=0.005$ following VWB19. We do not tune this parameter because, as you note, there is no fair way to do this. The results seem generally robust to the choice of q, at least so long as the network-structure term is upweighted relative to the predictive loss. We have added a line in Appendix F stating that we fix $q$ following previous work.
>
>
> Smaller points:
>
> We have added a sentence at line 276 clarifying that, in this simulation setup, $\psi$ is equal to $\beta_0$.
>
> We have added a sentence at line 205 to clarify that we use cross entropy for binary Y and MSE for continuous.

---

> > ### Author Response · Authors · 2022-08-08
> > **Questions about rebuttal**
> >
> > Thank you again for your review. Do you have any further questions?

---

### Official Review · Reviewer_muDH · 2022-07-14

**Rating:** 7
**Confidence:** 4
**Soundness:** 4 excellent
**Presentation:** 3 good
**Contribution:** 2 fair

**Summary:**

The authors propose a model and estimator for the "contagion" effect of treatments applied to a node's immediate neighbors in a graph structure. They control for homophily by conditioning on the node's graph embedding. Identifiability is formalized and the method is validated on semi-synthetic datasets.

**Questions:**

Please refer to the concerns above. I would be happy to amend my score if any of the major weaknesses are addressed in the rebuttal.

Maybe further expanding on the role of the aggregation function can illuminate the novelty of this approach.

**Limitations:**

I would suggest for the authors to briefly discuss the fundamental limitations in using a social-network structure to draw conclusions about social contagion in humans, which is really driven by many other factors.

**Strengths And Weaknesses:**

Strengths:
* The idea is clever and simple, yet highly relevant to the field of causal inference (on graphs in particular.)
* Most of the paper is clear and easy to read.
* The theorems are significant and their accompanying explanations are intuitive. This is especially the case for Theorem 2.

Weaknesses:
* I am a bit hesitant about the novelty of this work, as the method is quite similar to that introduced in [VWB19]. *Operationally*, the only major difference appears to be the aggregation operator that defines $V_i$ (Definition 1), which, coincidentally, is kept on rather vague terms until much later in the paper. The latter issue can be a source of confusion.
* The results are limited to synthetic outcome models. In the first experiment (6.1), the outcome's parametrization is linear with i.i.d Gaussian error, which matches the proposed loss function perfectly. The formulation of the second (6.2) is quite simple, which is compelling, but not a comprehensive evaluation. It would be helpful to explore the performance under minor model mismatch.


Smaller concerns:
* The `do` calculus should be introduced before using it in Equation 3.1. The notation $(T=t^*)$ should more clearly reflect the fact that it is a vector assignment. Also, the context preceding that equation is confusing. How can it be viewed as an intuitive choice for capturing the "average influence of a node's neighbors' treatments on its outcome" if there is no explicit incorporation of neighborhoods? That specification comes a whole section later, leaving the reader in the dark.
* The justification given in footnote #2 for considering the embeddings nonparametric is a bit of a stretch. Some distributional assumptions appear to be made implicitly, through the outer product and sigmoid in the loss function. Please clarify this claim.
* It would be helpful for the reader to clarify that $m_G$ is used to model a node's *potential outcomes*.
* In Section 7, the first paragraph on RCTs is outside the scope of this topic. Perhaps some of this space can be allocated for a conclusion section.

---

> ### Author Response · Authors · 2022-08-02
> **Rebuttal**
>
> Thank you for your thorough review and comments. We’re glad you found the paper clear, clever, and significant.
>
> * "I am a bit hesitant about the novelty of this work, as the method is quite similar to that introduced in [VWB19]. Operationally, the only major difference appears to be the aggregation operator that defines $V_i$ (Definition 1), which, coincidentally, is kept on rather vague terms until much later in the paper. The latter issue can be a source of confusion."
>
> The problem we address here---estimation of contagion effects---is fundamentally different from the problem addressed in VWB19---estimation of individual level effects without any contagion. We agree that in the actual implemented method, the operational difference is defining a surrogate treatment $V_i$ at each node $i$ via aggregation. However, we view this simplicity as a feature, not a bug! The novelty in this paper is giving a non-parametric formalization of the estimand and plausible conditions for identification. The novelty is not the method itself, but rather clarifying what it actually means as a causal estimation procedure. It is a fortunate coincidence that the actual method requires only a small technical modification relative to the simpler pure homophily case of VWB19.
>
> Regarding the aggregation variable $V_i$: we have added a footnote in Definition 1 to clarify that $S_Y$ is set by the analyst for the practical algorithm, as well as noting that we use the mean function throughout the paper.
>
> * "The results are limited to synthetic outcome models. In the first experiment (6.1), the outcome's parametrization is linear with i.i.d Gaussian error, which matches the proposed loss function perfectly. The formulation of the second (6.2) is quite simple, which is compelling, but not a comprehensive evaluation. It would be helpful to explore the performance under minor model mismatch."
>
> The use of synthetic data here is necessary because ground truth causal effects (required for evaluation) are not available in real-world data. Note, however, that the simulation setting is not crafted to match our model assumptions. In particular, the simulation relies on the ground truth value of the node attribute we’re using for simulation. The model has no access to this and, a priori, there is no reason to think that embedding methods should favor learning a representation of the latent feature that allows a linear map to predict Y. The relationship between the latent variable and the network structure is determined by the real world process that formed the graph. The fact that the method works is a consequence of the flexible modeling we use. (Put differently, if we instead had the confounder as $cos(g(C))$ or some other transformation of $C$, that would be exactly as well matched to the model setup). However, this would make it harder to explicitly control the degree of confounding in the simulation.
>
> Smaller concerns:
>
> * "The do calculus should be introduced before using it in Equation 3.1. The notation $do(T=t^*)$ should more clearly reflect the fact that it is a vector assignment. Also, the context preceding that equation is confusing. How can it be viewed as an intuitive choice for capturing the "average influence of a node's neighbors' treatments on its outcome" if there is no explicit incorporation of neighborhoods? That specification comes a whole section later, leaving the reader in the dark."
>
> In the 10-page paper version we will add the following clarifying text following equation 3.1: “Here, do is Pearl’s do notation, and indicates that the treatment (vector) of all nodes is intervened on and set to $t^*$. Note that, following the structural model in equation 2.1, nodes' outcomes are only affected by their neighbor’s treatments. Accordingly, the estimand is….”
>
> * "The justification given in footnote #2 for considering the embeddings nonparametric is a bit of a stretch. Some distributional assumptions appear to be made implicitly, through the outer product and sigmoid in the loss function. Please clarify this claim."
>
> In the 10-page paper version, we will remove the footnote and add the following more developed explanation in the paper body: “Although the map from the embeddings is linear, because the embeddings are unconstrained, the combined embedding-linear map can flexibly represent complex relationships between the network structure and outcome/treatment. In \cref{sec:experiments} we see that the model ably captures a relationship between Y and a latent attribute C that is non-linear in C, where the relationship between the network and C is complex.”

---

> > ### Author Response · Authors · 2022-08-08
> > **Questions about rebuttal**
> >
> > Thank you again for your review. Do you have any further questions?

---

> > ### Comment · Reviewer_muDH · 2022-08-08
> > **Response**
> >
> > Thank you for your explanations that address my major concerns. I am increasing my score accordingly.

---

### Author Response · Authors · 2022-08-02
**Response to reviewers**

We thank all reviewers for reading our paper and their thoughtful comments. All reviewers agreed that the paper addresses an important problem, and it is clear and well supported. Reviewers were concerned about the paper not providing a thorough empirical validation. We have addressed this by explaining that the main dataset used in the paper is only semi-synthetic (which is required because ground-truth peer contagion effects are not available in real world data), yet the network structure and relationship with hidden confounders are real and varied. Following reviewer suggestions, we have also added minor supplementary experiments (see Appendix) that show the method is robust to varying the noise and the unobserved confounding level parameters, as well as a demonstration on Wikipedia data showing that the method works on distinct types of network structure.

---

### Meta-Review · Area_Chair_MTtG · 2022-08-28

**Recommendation:** Accept
**Confidence:** Certain

**Metareview:**

This paper introduces a method for causal estimation of peer influence in networks in the presence of a confounding effect of homophily. The reviewers  agreed that the idea proposed in the paper is novel and well-motivated, and the accompanying theoretical analysis is useful. While there were concerns about the lack of experimental variation in more realistic scenarios, the consensus is that the contribution is sufficiently  novel and significant to warrant its acceptance.

**Award:**

No

---

### Decision · Program_Chairs · 2022-09-14

Accept